# Exploring the textural characteristics of foods preferred by Chinese elderly individuals based on IDDSI levels

**Muxi Chen**[1,2]☯*, **Yi Cheng**[1]☯*, **Wen Hu**[1], **Mengyan Wang**[1,2], **Juan Duan**[3], **Lei Shi**[1]*

**1** Department of Clinical Nutrition, West China Hospital, Sichuan University, Chengdu, China, **2** West China School of Clinical Medicine, Sichuan University, Chengdu, China, **3** School of Public Health, Sichuan University, Chengdu, China

☯ These authors contributed equally to this work.
* Shileiaini_123@163.com (LS); 3255632588@qq.com (MC); Miranda1024@wchscu.cn (YC)

## Abstract

**Data Availability Statement:** All relevant data are within the paper and its Supporting Information files.

### Objective

This study aimed to investigate the textural characteristics of foods preferred by elderly Chinese individuals and their suitability based on the International Dysphagia Diet Standardization Initiative (IDDSI) framework. The goal was to provide objective data to support the development of safe and nutritious diets tailored to the swallowing abilities of the elderly.

### Methods

A cross-sectional observational study was conducted, using web-scraping technology to identify 26 commonly preferred food ingredients among elderly individuals across seven regions of China. These foods were prepared and evaluated according to IDDSI levels 0–7. Texture analysis was performed to measure hardness, cohesiveness, and adhesiveness, with statistical tests including chi-square analysis and multiple linear regression used to explore the relationships between these textural properties and IDDSI levels.

### Results

As IDDSI levels increased, the hardness of various food categories generally showed an upward trend, with significant increases observed in fruits, vegetables, grains, and tubers at IDDSI levels 6–7 ($p \leq 0.05$). Cohesiveness varied without a clear linear trend, showing significant changes at specific IDDSI levels for meats, grains, and tubers ($p \leq 0.05$). Adhesiveness initially increased and then decreased across the IDDSI spectrum, with the most significant fluctuations occurring in mixed beans, fruits, and grains at levels 5–7 ($p \leq 0.05$). Regression analysis revealed that cohesiveness had the most substantial impact on IDDSI levels (coefficient = -5.224, $p \leq 0.05$), followed by adhesiveness (coefficient = -0.021, $p \leq 0.05$), and hardness (coefficient = 0.002, $p \leq 0.05$).

**Funding:** 2023YFF1104405 老年人群营养健康食品创制和产业化示范

**Competing interests:** The authors have declared that no competing interests exist.

## Conclusion

The findings underscore the importance of considering the textural properties of foods when designing diets for elderly individuals with dysphagia. The study provides empirical evidence supporting the IDDSI framework and offers a predictive model that can assist in the development of customized diets, ensuring both safety and nutritional adequacy. Future research should focus on refining food preparation methods to better meet the specific needs of this population.

## 1. Introduction

As China has an aging society, by the end of 2022, the population aged 60 and above accounted for 19.8% of the total population, an increase of 1.1% compared to the seventh national census in 2020 [1]. With physiological aging, swallowing function begins to deteriorate, and common chronic conditions, such as diabetes, cardiovascular disease, and neurological disorders, gradually lead to functional impairments [2]. Studies have shown that the prevalence of dysphagia among Chinese elderly individuals is 66.0%, with 21.0% in those aged 60–69, 28.0% in those aged 70–79, and 41.0% in those aged 80 and above. The incidence of swallowing dysfunction is 50%-60% after head and neck cancer treatment and up to 37%-78% after stroke [2, 3]. Dysphagia, which worsens with aging or disease progression, changes the dietary status, nutritional status, and disease outcomes of elderly individuals, reducing their quality of life and potentially causing more severe physical and mental health issues [4, 5]. Therefore, addressing the dietary needs of elderly individuals with varying degrees of swallowing ability and providing safe, nutritious, and suitable diets are crucial issues for successful aging and home-based elderly care.

Given the diversity of global diets and the multiple methods of food cognition and evaluation, the International Dysphagia Diet Standardization Initiative (IDDSI) was formed in 2013 by an interdisciplinary team of healthcare professionals, scholars, industry representatives, and patient representatives from multiple countries. In 2016, the IDDSI proposed a standardized framework for texture-modified foods and thickened liquids using globally unified, standardized operations and sensory evaluations to assist in the appropriate modification of dysphagia diets [6]. However, despite the unified evaluation and operational guidelines of the IDDSI, its evaluation methods are based on subjective perceptions and lack objective data support. In practical applications, this approach does not answer questions about the selection and preparation of dysphagia diets in complex swallowing disorder situations.

Considering the diversity of Chinese dietary ingredients and regions, this study is the first to use web-scraping technology to screen for food ingredients preferred by Chinese elderly people and to prepare these ingredients in corresponding forms based on IDDSI levels of 0–7. Texture measurements were also conducted to support the use of the IDDSI in sensory evaluations with objective data, aiming to address the lack of objectivity in existing standards and to provide a theoretical basis and research evidence for designing medical dietary menus suitable for Chinese elderly individuals at different IDDSI levels.

## 2. Methods

This study was conducted as a cross-sectional observational study aimed at exploring the texture characteristics of foods commonly preferred by elderly individuals in China, using the IDDSI framework as a guide.

## 2.1 Food data screening

To gain a deep understanding of the dietary preferences of Chinese elderly people, this study used the Python programming language and its web-scraping library (Scrapy) to extract data from the Chinese cuisine website "Meishichina" (www.meishichina.com), covering seven major regions and the recommended recipes for elderly people. In compliance with the website's access policy and relevant laws and regulations, the ingredient information was extracted from these websites. The data were then cleaned and organized to remove duplicates and irrelevant content, and only key information was retained for subsequent analysis.

## 2.2 Selection of experimental materials

Using web scraping technology, this study selected 26 of the most common and recommended food ingredients for elderly people from Chinese culinary websites across seven regions. These ingredients include meat (pork, beef, chicken, fish), eggs (chicken eggs), grains and tubers (noodles, rice, oats, potatoes, sweet potatoes, yams, pumpkins), vegetables (tomatoes, eggplants, cabbages, carrots, greens, broccoli, spinach), mixed beans (red beans, green beans), fruits (apples, pears, peaches, bananas), and tofu.

## 2.3 Food processing methods

The selected ingredients were processed using typical daily cooking methods. Preparation generally involved washing, cutting into pieces, and steaming until fully cooked. After steaming, the ingredients were blended into a puree, and different amounts of water were added as needed to achieve the desired consistency and softness. For example, "pumpkin puree (1:2)" indicates a pumpkin-to-water ratio of 1:2. All subsequent examples follow this notation without further explanation.

The preparation methods for each food category were as follows:

Meat Samples: Fresh meat was trimmed of sinew and fat, cut into pieces (approximately 2 cm × 2 cm), washed, ground into a paste, steamed for 30 minutes, and then mixed with various amounts of water to produce different textures (e.g., pork puree (1:0)).

Eggs: Hard-boiled eggs were separated, and the yolks were blended with water to achieve different consistencies (e.g., egg yolk (1:0.5)). For steamed eggs, beaten eggs were mixed with water and then steamed (e.g., steamed egg (1:2)).

Grains and Tubers: Rice was soaked for 30 minutes and then steamed with a certain amount of water (e.g., rice (1:4)). Oats were cooked with water (e.g., oats (1:8)). Noodles were boiled for 8 minutes, ground into a paste, and mixed with water (e.g., noodles (1:1)).

Vegetables: Vegetables were washed, cut into pieces (2 cm × 2 cm), steamed for 30 minutes, and then blended with water into a puree (e.g., carrot puree (1:0.5)).

Mixed Beans: Beans were soaked for 12 hours, steamed for 30 minutes, and then blended with water (e.g., red bean puree (1:2)).

Fruits: Fruits were washed, peeled, cored, diced (2 cm × 2 cm), and blended with water or first steamed and then blended (e.g., apple puree (1:1)) [7].

Uniform processing steps were followed to ensure consistency across samples. Three samples (each weighing approximately 50 g) of each food in different processed states were prepared for subsequent IDDSI level evaluation and texture measurements.

## 2.4 IDDSI evaluation methods

An IDDSI evaluation toolkit, including a spoon, fork, and syringe, was prepared for IDDSI assessment. In a quiet environment with appropriate lighting, the fluidity of the liquids was

tested using a syringe. The syringe was filled with 10 mL of the liquid sample, a finger was placed over the nozzle to stop the flow, and the amount of liquid remaining in the syringe after 10 seconds was recorded.

IDDSI Level 0: All liquid flows through the syringe.

IDDSI Level 1: 1 mL-4 mL remains.

IDDSI Level 2: 4 mL-8 mL remains.

IDDSI Level 3: 8 mL-10 mL remains.

IDDSI Level 4: No fluid flows after 10 seconds.

For samples that did not flow through the syringe, a fork test was used. The standard metal fork has a gap between the tines of 4 mm and a total width of 1.5 cm. The food was evaluated to determine whether it could be crushed by the fork, hung on the fork, and so on to assess its viscosity and cohesiveness.

IDDSI Level 5: Food easily separates and flows between the fork tines.

IDDSI Level 6: Food does not flow between the fork tines.

IDDSI Level 7: Food can be easily separated by the edge of a fork or spoon.

For IDDSI Levels 6–7, an additional spoon test was used to evaluate whether the food could be easily lifted by a spoon and whether it maintained its shape without flowing. A finger pressure test was also conducted by lightly pressing the surface of the food to assess its elasticity and firmness. Each processed food sample was tested according to the above procedures, and the food was assigned to the corresponding IDDSI level based on the test results. For borderline cases, adjustments were made based on actual conditions and needs [6].

## 2.5 Texture measurement methods

A texture analyzer TA.new plus (ISENSO, USA) was used to measure the texture of each food sample, including hardness, cohesiveness, and adhesiveness. Texture profile analysis was selected, with the test mode set to compression, the target mode set to distance, and a target value of 3.000 mm. The pretest and posttest speeds were set to 0.50 mm/s, with the trigger mode set to Force and the trigger point set at 3.000 gf. The data were recorded after each test, and the average texture data for each food were calculated from the three samples.

## 2.6 Statistical analysis methods

Frequencies and percentages were used to describe the distribution of food categories at different IDDSI levels. Chi-square analysis was used to determine if the differences in the distribution of food categories at the IDDSI levels were statistically significant. For significant chi-square results, pairwise comparisons were conducted to identify specific IDDSI levels with differing food category distributions. Paired t tests were used to analyze the changes in textural properties (hardness, cohesiveness, adhesiveness) of foods at different IDDSI levels, comparing the average differences of the same food category at adjacent IDDSI levels to determine if textural changes were statistically significant. Multiple linear regression models were used to analyze the impact of hardness, cohesiveness, and adhesiveness on IDDSI levels. After Z score outlier processing (excluding data points with absolute Z scores greater than 3), the dataset included 556 samples. Statistical analysis was performed using SPSS (25.0) software, with a significance level set at $p \leq 0.05$ for all tests.

Frequencies and percentages were used to describe the distribution of food categories across different IDDSI levels. Chi-square tests were applied to assess whether the differences in the distribution of food categories among IDDSI levels were statistically significant. When significant results were found, pairwise comparisons were performed to pinpoint specific IDDSI levels with differing food category distributions.

For the analysis of textural properties (hardness, cohesiveness, adhesiveness) across different IDDSI levels, we initially considered using paired t-tests. However, the results of the Shapiro-Wilk test indicated that none of the datasets followed a normal distribution ($p$-values were all significantly below 0.05, with W Statistics showing significant deviations from normality: Adhesiveness: W = 0.6342, $p < 0.001$; Hardness: W = 0.3223, $p < 0.01$; Cohesiveness: W = 0.9716, $p < 0.01$). As a result, we revised our statistical approach and replaced the paired t-test with non-parametric alternatives. Specifically, we used the Mann-Whitney U test to compare the textural properties between adjacent IDDSI levels within the same food category, allowing us to determine if the changes were statistically significant.

Multiple linear regression models were employed to analyze the influence of hardness, cohesiveness, and adhesiveness on the IDDSI levels. After processing outliers using Z scores (excluding data points with absolute Z scores greater than 3), the dataset included 556 samples. All statistical analyses were conducted using SPSS (version 25.0), with the significance level set at $p \leq 0.05$ for all tests.

## 3. Results

### 3.1 Distribution of different food categories at each IDDSI level

At IDDSI level 0, fruits accounted for the greatest proportion (31.00%), followed by vegetables (24.14%). At Level 1, vegetables had the highest proportion (45.45%), followed by grains and tubers (31.82%). At Level 2, vegetables again led in proportion (23.81%), followed equally by fruits and grains and tubers (19.05% each). At Level 3, grains and tubers were the most common (38.10%), followed by vegetables (28.57%). At Level 4, vegetables were again at the top (35.71%), followed by grains and tubers (28.57%). At Level 5, the percentages of grains and tubers were the highest (35.00%), followed by those of meats and mixed beans (20.00% each). At Level 6, grains and tubers were most prevalent (42.86%), followed by vegetables (28.57%). At Level 7, the proportions of meats, grains and tubers, eggs, and mixed beans were the same (25.00%).

The chi-square test analysis of the distribution differences of various ingredients across IDDSI levels indicated statistically significant differences at IDDSI level 0 ($\chi^2 = 47.97$, $p \leq 0.05$), IDDSI level 1 ($\chi^2 = 27.48$, $p \leq 0.05$), IDDSI level 5 ($\chi^2 = 13.98$, $p \leq 0.05$), IDDSI level 6 ($\chi^2 = 15.98$, $p \leq 0.05$), and IDDSI level 7 ($\chi^2 = 18.80$, $p \leq 0.05$). For IDDSI Level 2 ($\chi^2 = 9.95$, $p = 0.127 > 0.05$), Level 3 ($\chi^2 = 9.87$, $p = 0.13 > 0.05$), and Level 4 ($\chi^2 = 8.91$, $p = 0.179 > 0.05$), the distribution differences were not statistically significant, indicating a varied significance in the distribution of food categories across IDDSI levels (Table 1).

For each food category, fruits had the highest proportion at IDDSI Level 0 (42.86%); vegetables were most prevalent at IDDSI Level 1 (24.39%); meats reached their peak at IDDSI Level 0 (31.25%), with a secondary peak at Level 5 (25.00%); grains and tubers were most common at IDDSI Level 3 (20.00%), followed by levels 1 and 5 (17.5% each); eggs dominated at IDDSI Level 2 (30.00%), with subsequent highs at levels 0 and 4 (20.00% each); tofu was most frequent at IDDSI Level 0 (50.00%); and mixed beans shared their highest proportion at levels 2, 3, and 6 (18.18% each).

In this study, the chi-square test was employed to analyze the distribution variance of each food category across different IDDSI levels. Significant differences were observed in the distributions of vegetables ($\chi^2 = 38.05$, $p \leq 0.05$), meats ($\chi^2 = 34.29$, $p \leq 0.05$), grains and tubers ($\chi^2 = 27.43$, $p \leq 0.05$), eggs ($\chi^2 = 20.57$, $p \leq 0.05$), and tofu ($\chi^2 = 34.29$, $p \leq 0.05$) across IDDSI levels, indicating significant variance. The mixed bean variable did not significantly differ across the different IDDSI levels ($\chi^2 = 4.68$, $p = 0.700 > 0.05$) (Table 2).

**Table 1. Distribution of different ingredients at the same IDDSI level (Percentage and frequency) and statistical significance analysis (Chi-square test)\*.**

| IDDSI Level/Food Category | Fruits(%)/N | Vegetables(%)/N | Meats(%)/N | Grains and Tubers(%)/N | Eggs(%)/N | Tofu(%)/N | Mixed Beans(%)/N |
|---|---|---|---|---|---|---|---|
| Level 0* | 31.03(27) | 24.14(21) | 17.24(15) | 10.34(9) | 6.90(6) | 6.90(9) | 3.45(3) |
| Level 1* | 18.18(12) | 45.45(30) | 0.00(0) | 31.82(21) | 0.00(0) | 0.00(0) | 4.55(3) |
| Level 2 | 19.05(12) | 23.81(15) | 9.52(6) | 19.05(12) | 14.29(9) | 4.76(3) | 9.52(6) |
| Level 3 | 9.52(6) | 28.57(18) | 9.52(6) | 38.10(24) | 0.00(0) | 4.76(3) | 9.52(6) |
| Level 4 | 7.14(3) | 35.71(15) | 7.14(3) | 28.57(12) | 14.29(6) | 0.00(0) | 7.14(3) |
| Level 5* | 5.00(3) | 20.00(12) | 20.00(12) | 35.00(21) | 5.00(3) | 5.00(3) | 10.00(3) |
| Level 6* | 0.00(0) | 28.57(12) | 7.14(3) | 42.86(18) | 7.14(3) | 0.00(0) | 14.29(6) |
| Level 7* | 0.00(0) | 0.00(0) | 25.00(3) | 25.00(3) | 25.00(3) | 0.00(0) | 25.00(3) |

\* Indicates a statistically significant *p* value

## 3.2 Texture data of ingredients at different IDDSI levels

**3.2.1 Hardness of different food categories at IDDSI levels.**  As the IDDSI levels increased, the hardness of various food categories generally showed an increasing trend, with most categories reaching their maximum hardness at Level 7, particularly in fruits. Specifically, fruits, vegetables, grains and tubers, and tofu exhibited significant increases in hardness at IDDSI levels 6–7 ($p \leq 0.05$). Notably, fruits experienced the highest increase in hardness. Eggs showed a significant increase in hardness at levels 6–7, while mixed beans and meats also displayed notable increases at levels 6–7 ($p > 0.05$). The range of hardness across different IDDSI levels fluctuated to varying degrees, with the most notable fluctuations occurring in meats at Level 5, in eggs, mixed beans, and tofu at Level 6, and in fruits, vegetables, and grains and tubers at Level 7, which exhibited the largest fluctuation range (Fig 1 and Table 3).

**3.2.2 Cohesiveness of different food categories at IDDSI levels.**  The cohesiveness range of various foods across different IDDSI levels shows considerable variability. Overall, the cohesiveness values fluctuate with changes in IDDSI levels, without following a clear linear trend. At IDDSI Levels 2–3, meats and mixed beans exhibit a higher cohesiveness, whereas vegetables demonstrate greater cohesiveness at lower levels like 0–1. At IDDSI Levels 3–4, grains and tubers display an increase in cohesiveness, and fruits show significant cohesiveness changes at Levels 5–6.

Moreover, for the same food category, the most significant change in cohesiveness is observed for meats at IDDSI Levels 2–3, mixed beans at Levels 2–3, grains and tubers at Levels 3–4, vegetables at Levels 0–1, and fruits at Levels 5–6, all with statistical significance ($p \leq 0.05$).

**Table 2. Distribution of the same ingredient across different IDDSI levels (Percentages and frequencies) and statistical significance analysis (Chi-square analysis).**

| IDDSI Level/Food Category | Fruits(%)\*/N | Vegetable(%)\*/N | Meats(%)\*/N | Grains and Tubers(%)\*/N | Eggs(%)\*/N | Tofu(%)\*/N | Mixed Beans(%)/N |
|---|---|---|---|---|---|---|---|
| Level 0 | 42.86(27) | 17.07(21) | 31.25(15) | 7.50(9) | 20.00(6) | 50.00(9) | 9.09(3) |
| Level 1 | 19.05(12) | 24.39(30) | 0.00(0) | 17.50(21) | 0.00(0) | 0.00(0) | 9.09(3) |
| Level 2 | 19.05(12) | 12.20(15) | 12.50(6) | 10.00(12) | 30.00(9) | 16.67(3) | 18.18(6) |
| Level 3 | 9.52(6) | 14.63(18) | 12.50(6) | 20.00(24) | 0.00(0) | 16.67(3) | 18.18(6) |
| Level 4 | 4.76(3) | 12.20(15) | 6.25(3) | 10.00(12) | 20.00(6) | 0.00(0) | 9.09(3) |
| Level 5 | 4.76(3) | 9.76(12) | 25.00(12) | 17.50(21) | 10.00(3) | 16.67(3) | 9.09(3) |
| Level 6 | 0.00(0) | 9.76(12) | 6.25(3) | 15.00(18) | 10.00(3) | 0.00(0) | 18.18(6) |
| Level 7 | 0.00(0) | 0.00(0) | 6.25(3) | 2.50(3) | 10.00(3) | 0.00(0) | 9.09(3) |

\* Indicates a statistically significant *p* value

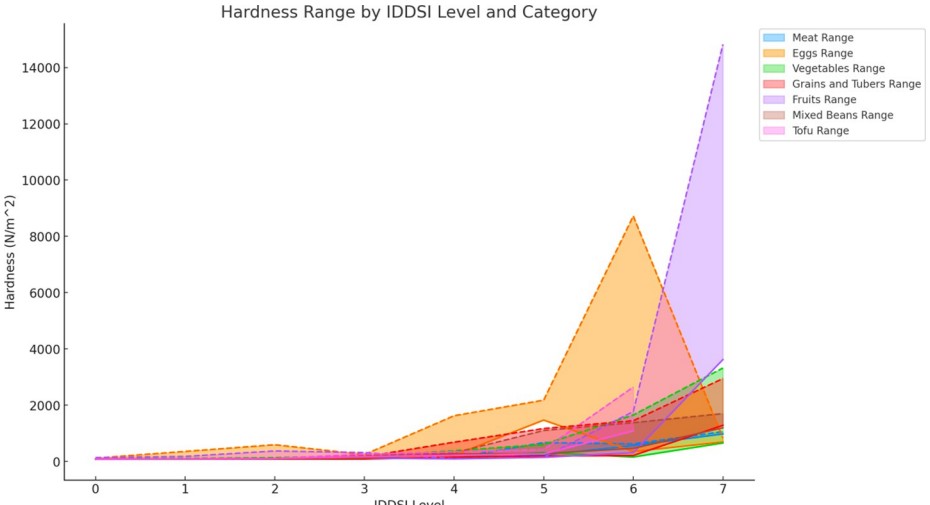

**Fig 1. Comparison of hardness ranges for different food categories under IDDSI levels.**

Eggs also show significant changes in cohesiveness at IDDSI Levels 6–7 ($p \leq 0.05$). However, the change in cohesiveness for tofu at IDDSI Levels 1–2, although observed with a Δ of 0.268 gf, does not have statistical significance ($p > 0.05$).

The cohesiveness range of various food categories fluctuates differently across IDDSI levels, with tofu at IDDSI Levels 1–2, fruits and meats at Level 5, vegetables and mixed beans at Level 3, and grains and tubers at Level 6 showing the most significant fluctuations (Fig 2 and Table 4).

**3.2.3 Adhesiveness of different food categories at IDDSI levels.** Overall, as IDDSI levels increase, the adhesiveness of most food categories shows varying trends, with some categories exhibiting significant increases followed by decreases, while others maintain more consistent fluctuations. This is particularly evident between IDDSI levels 3 to 7, where food categories such as mixed beans, fruits, and grains and tubers demonstrate significant changes in adhesiveness. Among these, mixed beans show the most pronounced increase at IDDSI Levels 5–6 with a maximum adhesiveness Δ of 89.280 gf, and both fruits and grains and tubers peak at Levels 3–4 and 6–7 with maximum adhesiveness Δ values of 25.933 gf and 39.138 gf, respectively. These changes are statistically significant ($p \leq 0.05$). Additionally, tofu exhibits a significant change at IDDSI Levels 2–3, with a maximum adhesiveness Δ of 18.318 gf, also

**Table 3. Hardness characteristics and statistical significance analysis of different ingredients across IDDSI levels*.**

| Food Category | IDDSI Level for Max Hardness Δ | Max Hardness Δ (N/m^2) | IDDSI Level for Max Hardness Fluctuation Range | Max Hardness Fluctuation Range (N/m^2) (Range) |
|---|---|---|---|---|
| Meats | 6–7 | 430.441 | 5 | 442.463 |
| Eggs | 6–7 | 2473.320 | 6 | 8370.388 |
| Vegetable | 6–7 | 822.204* | 7 | 2664.945 |
| Grains and Tubers | 6–7 | 1514.951* | 7 | 1661.232 |
| Fruits | 6–7 | 7142.542* | 7 | 11201.329 |
| Mixed Beans | 6–7 | 785.209 | 6 | 910.873 |
| Tofu | 5–6 | 1476.725* | 6 | 1558.168 |

* Indicates a statistically significant *p* value

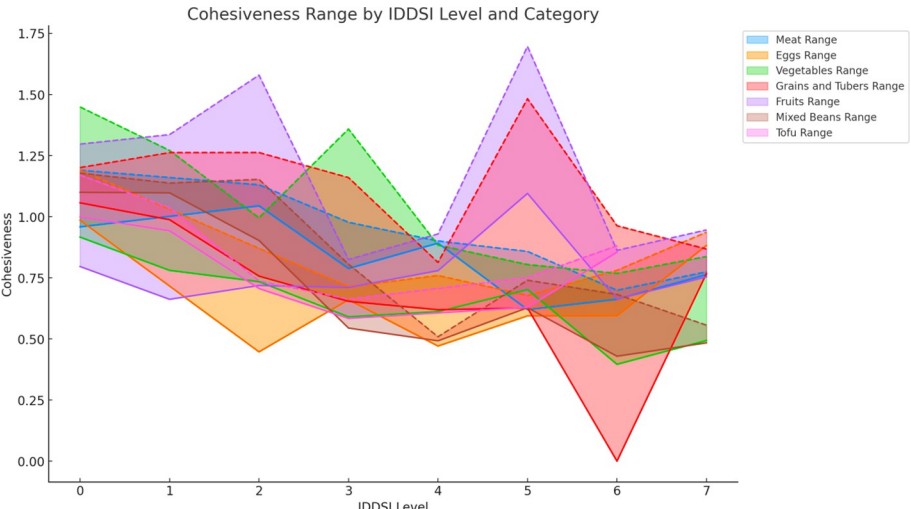

**Fig 2. Comparison of the cohesion ranges of different food categories under IDDSI levels.**

statistically significant. In contrast, although meats and eggs reach their maximum adhesiveness Δ values at IDDSI Levels 6–7 and 5–6, with 19.790 gf and 14.380 gf respectively, these changes do not achieve statistical significance and therefore lack statistical representation.

The fluctuation range of adhesiveness varies across different food categories. Fruits at IDDSI Level 3, vegetables at Level 4, mixed beans and meats at Level 5, and grains and tubers at Level 6 experience the most significant fluctuations (largest range) in adhesiveness (Fig 3 and Table 5).

## 3.3 Relationships between IDDSI levels and texture characteristics (Hardness, cohesiveness, adhesiveness)

**3.3.1 Visual analysis: Scatter plots.**   Scatter plots were used to visualize the relationships between the IDDSI and hardness, cohesiveness, and adhesiveness. As shown in the figures, the relationship between hardness and the IDDSI is evident. Hardness varied significantly between different IDDSI levels, especially in the mid-to-low range, with an increase in hardness associated with higher IDDSI levels (Fig 4). As the IDDSI increases, cohesiveness generally shows a slight decreasing trend. At IDDSI Level 4, the cohesiveness values tended to stabilize and

**Table 4. Cohesion characteristics and statistical significance analysis of different ingredients across IDDSI levels\*.**

| Food Category | IDDSI Level for Max Cohesiveness Δ | Max Cohesiveness Δ Change (gf) | IDDSI Level for Max Cohesiveness Fluctuation Range | Max Cohesiveness Fluctuation Range (gf) |
|---|---|---|---|---|
| Meats | 2–3 | 0.209* | 5 | 0.861 |
| Eggs | 6–7 | 0.233* | 2 | 0.424 |
| Vegetable | 0–1 | 0.125* | 3 | 0.768 |
| Grains and Tubers | 3–4 | 0.214* | 6 | 0.963 |
| Fruits | 5–6 | 0.557* | 2 | 0.861 |
| Mixed Beans | 2–3 | 0.374* | 3 | 0.261 |
| Tofu | 1–2 | 0.268 | 0 | 0.171 |

\* Indicates a statistically significant *p* value

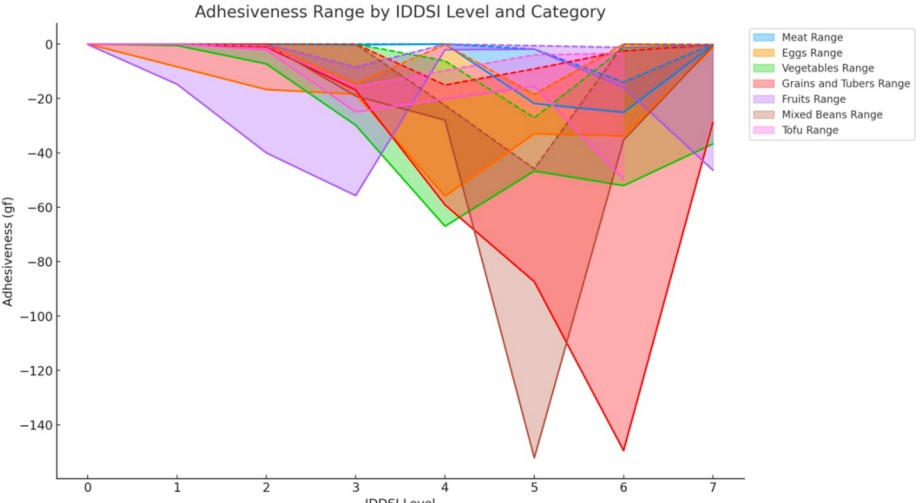

**Fig 3. Comparison of adhesiveness ranges for different food categories under IDDSI levels.**

remain relatively consistent from IDDSI Levels 5 to 7 (Fig 5). Adhesiveness (absolute value) initially increases and then decreases as the IDDSI increases. From IDDSI Levels 1 to 4, the increase in adhesiveness (absolute value) was significant, while from IDDSI Levels 5 to 7, there was a more moderate decreasing trend (Figs 4–6).

**3.3.2 Multiple linear regression analysis: Impact of food texture characteristics on IDDSI levels.** A multiple linear regression analysis was conducted to determine the relative influence of texture characteristics—adhesiveness, cohesiveness, and hardness—on IDDSI levels. These values were derived from the intercept and coefficients estimated in the multiple regression model (Table 6). The results indicate that cohesiveness (in gf) has the most significant impact on IDDSI levels, with a coefficient of -5.224 ($p \leq 0.05$). Adhesiveness (in gf) also significantly influenced the IDDSI levels, showing a coefficient of -0.021 ($p \leq 0.05$). Hardness (in N/m$^2$) had a smaller but still significant effect, with a coefficient of 0.002 ($p \leq 0.05$). The model's R-squared value of 0.695 and adjusted R-squared value of 0.694 suggest a strong explanatory power (Table 6).

**Table 5. Adhesion characteristics and statistical significance analysis of different ingredients across IDDSI levels\*.**

| Food Category | IDDSI Level for Max Adhesiveness Δ | Max Adhesiveness Δ (gf) | IDDSI Level for Max Adhesiveness Fluctuation Range | Max Adhesiveness Fluctuation Range Δ (gf) |
|---|---|---|---|---|
| Meats | 6–7 | 19.790 | 5 | 20.030 |
| Eggs | 5–6 | 14.380 | 4 | 55.820 |
| Vegetable | 3–4 | 19.555* | 4 | 60.840 |
| Grains and Tubers | 6–7 | 39.138* | 6 | 147.010 |
| Fruits | 3–4 | 25.933* | 3 | 47.260 |
| Mixed Beans | 5–6 | 89.280* | 5 | 106.290 |
| Tofu | 2–3 | 18.318* | 6 | 46.520 |

\* Indicates a statistically significant *P* value

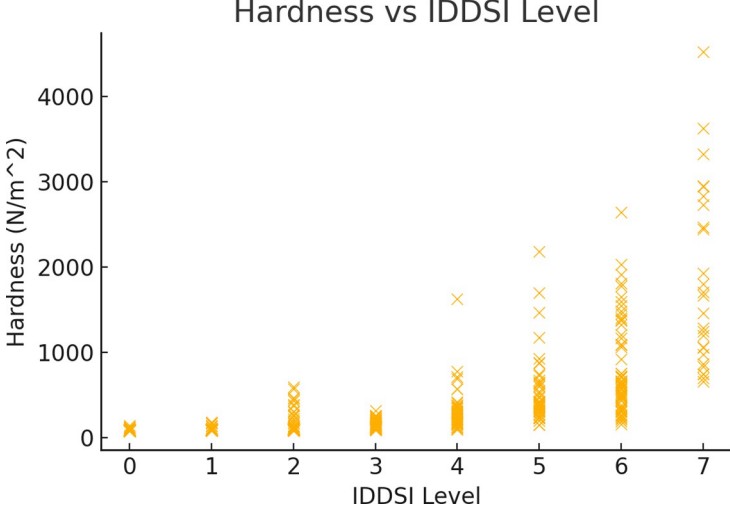

**Fig 4. Relationships between hardness and IDDSI levels.**

## 4. Discussion

### 4.1 Texture characteristics of preferred ingredients among elderly Chinese individuals based on IDDSI levels

This study explored the textural characteristics of foods preferred by elderly Chinese individuals, focusing on how these characteristics align with the International Dysphagia Diet Standardization Initiative (IDDSI) levels. The results demonstrated that hardness, cohesiveness, and adhesiveness of foods varied across different IDDSI levels, with some food categories exhibiting significant changes that warrant closer examination [7]. Hardness was found to generally increase with higher IDDSI levels, which is consistent with the IDDSI framework's expectation that more solid foods require greater oral strength. Notably, fruits showed the most significant increase in hardness at IDDSI level 7, suggesting that extra care should be taken in preparing fruit-based foods for individuals with reduced chewing ability. Studies have

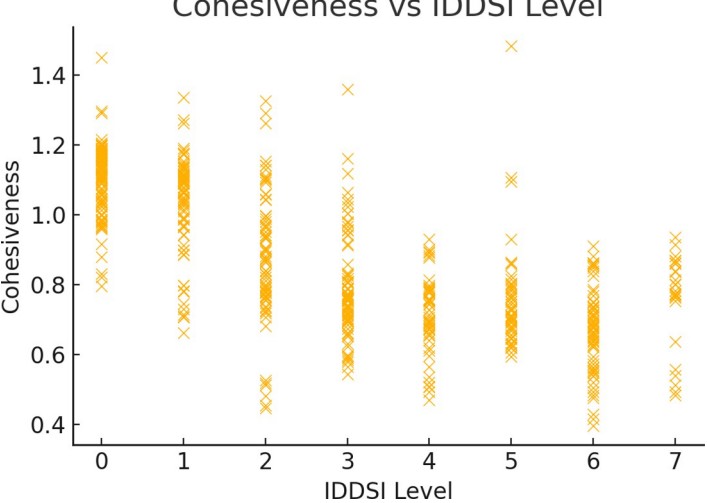

**Fig 5. Relationships between cohesiveness and IDDSI levels.**

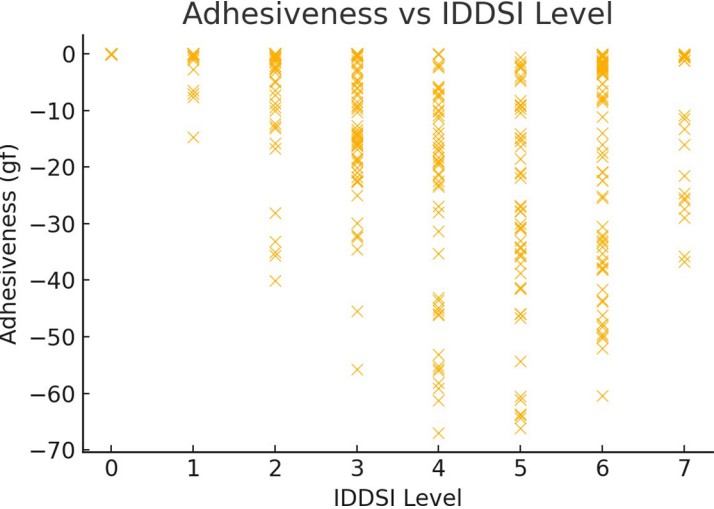

**Fig 6. Relationships between adhesiveness and IDDSI levels.**

observed that texture-modified foods prepared for dysphagia patients exhibit similar trends in hardness [8]. Cohesiveness did not follow a clear linear trend, which reflects the complexity of food processing and preparation. For instance, vegetables and fruits demonstrated higher cohesiveness at lower IDDSI levels but showed significant variability at specific levels, such as IDDSI level 6 for vegetables. This variability suggests that food texture is influenced by multiple factors, including preparation methods and the physical properties of the food itself [9].

## 4.2 Relationships between IDDSI levels and hardness, cohesiveness, and adhesiveness

The relationship between IDDSI levels and food texture characteristics such as hardness, cohesiveness, and adhesiveness is well-documented in dysphagia research. Hardness is the force required by teeth to chew food and can directly impact the formation of a food bolus during eating for individuals with dysphagia [10]. This study revealed that as the IDDSI score increased, the hardness of food also increased, which aligns with the basic expectations of the IDDSI framework [6]. An increase in hardness indicates that eaters need more oral and pharyngeal muscle strength to handle foods with higher IDDSI levels [11]. Previous research supports these findings, emphasizing the importance of tailored food preparation methods to meet the specific hardness requirements of different IDDSI levels [12].

Cohesiveness is the ability of food to maintain its structural integrity during chewing and swallowing, serving as an important indicator for assessing swallowing risk [13]. This study

**Table 6. Multiple linear regression analysis results.**

| Variable | Coefficient | Standard Error | t value | P value | 95% Confidence Interval |
|---|---|---|---|---|---|
| Intercept | 6.562 | 0.281 | 23.344 | $\leq 0.05$ | (6.010, 7.115) |
| Adhesiveness (gf) | -0.021 | 0.004 | -5.587 | $\leq 0.05$ | (-0.029, -0.014) |
| Cohesiveness(gf) | -5.224 | 0.287 | -18.227 | $\leq 0.05$ | (-5.787, -4.661) |
| Hardness (N/m$^2$) | 0.002 | 0.000 | 16.814 | $\leq 0.05$ | (0.002, 0.002) |

Note: Other statistical indicators: R-squared: 0.695; Adj. R-squared: 0.694; F-statistic: 419.8; Prob (F-statistic): 5.66e-142; Durbin-Watson: 0.513; Condition Number: 4780; AIC: 1803; BIC: 1820.

revealed that as the IDDSI increased, cohesiveness generally decreased and stabilized above IDDSI level 4. The decreasing trend in cohesiveness reflects a reduction in the resistance to structural damage as the IDDSI increases, indicating that foods with higher IDDSI levels are more likely to be prepared as soft solids, increasing structural integrity and reducing water content or increasing structural components such as proteins and dietary fibers [14, 15].

Adhesiveness is the force required to overcome food sticking to the mouth during chewing and swallowing [16]. The study revealed that adhesiveness (absolute value) initially increased with increasing IDDSI and then decreased, starting to weaken gradually after IDDSI level 4. This phenomenon may be related to the physical changes in food transitioning from soft to more solid forms. At lower levels, the density of food texture increases with IDDSI levels, thus increasing adhesiveness. As the IDDSI level continues to rise and food transitions from liquid to solid, it becomes increasingly harder, making the surface less sticky, thereby reducing adhesiveness. Additionally, studies have shown that adhesiveness can be influenced by various factors, such as water content, processing methods, and food temperature [17]. Increasing the water content can significantly enhance adhesiveness up to a certain threshold, after which it begins to decrease [18].

This study used multiple linear regression analysis to quantify the relationships between food hardness, cohesiveness, and adhesiveness and IDDSI levels. The results showed that an increase in hardness was significantly associated with higher IDDSI levels, with a coefficient of 0.002, aligning with the physical requirements of food at different IDDSI levels. Cohesiveness had the most significant impact on IDDSI levels, with a coefficient of -5.224, indicating that an increase in cohesiveness significantly decreases the IDDSI level. This reflects that in food texture, increased cohesiveness means that food is easier to process and swallow and is suitable for patients requiring lower IDDSI levels. Adhesiveness also significantly affected IDDSI levels, with a coefficient of -0.021, suggesting that increased adhesiveness slightly lowers the IDDSI level, indicating that for foods with lower IDDSI levels, more attention should be given to residual food in the mouth [15, 19]. Therefore, the predictive equation for the relationship between the IDDSI and texture characteristics is:

"IDDSI Level = 6.562+(−0.021*Adhesiveness)+(−5.224*Cohesiveness)+(0.002*Hardness".

This model can explain approximately 69.4% of the variance and can be used to estimate the expected IDDSI level given specific texture parameters. By inputting the adhesiveness, cohesiveness, and hardness values of a particular food, the IDDSI can be calculated to aid decision-makers in the food manufacturing and healthcare sectors in making appropriate food choices and dietary arrangements.

## 4.3 Strengths, limitations and future recommendations

This study presents several strengths that contribute to the growing body of knowledge on dysphagia management. First, the use of web-scraping technology to gather data on food preferences among elderly Chinese individuals provides a novel approach to understanding dietary habits within this population. This method allowed for the inclusion of a wide range of food types, ensuring a comprehensive analysis of the textural characteristics aligned with IDDSI levels. Additionally, the application of multiple linear regression models to analyze the impact of food texture characteristics on IDDSI levels offers a quantitative approach that strengthens the study's findings.

However, several limitations should be noted. The study relied on data from a single source, the Chinese cuisine website "Meishichina," which may not fully represent the diversity of food preferences across different regions in China. Furthermore, the study's cross-sectional design limits its ability to establish causality between food texture characteristics and swallowing

safety. Finally, the use of laboratory-based texture measurements may not fully capture the complexities of real-world food preparation and consumption, potentially limiting the generalizability of the findings to everyday settings.

Based on the findings of this study, several future research directions are recommended. First, expanding the data sources to include a more diverse range of culinary websites and regional food preferences would provide a more comprehensive understanding of the dietary habits of elderly individuals across China. Longitudinal studies should also be conducted to explore the causal relationships between food texture characteristics and swallowing safety over time. Additionally, future research could focus on developing and testing food preparation methods that are specifically tailored to meet the IDDSI level requirements while maintaining the nutritional value and cultural acceptability of the foods. Finally, incorporating real-world testing with actual dysphagia patients in both clinical and home settings would enhance the external validity of the findings and provide more actionable insights for practitioners.

## 5. Conclusion

In summary, this study explored the texture characteristics of preferred ingredients among elderly Chinese individuals based on IDDSI levels, revealing how hardness, cohesiveness, and adhesiveness change with IDDSI levels. Experimental analysis showed that as IDDSI levels increase, food hardness generally increases, with fruits showing the most significant increase. The changes in cohesiveness and adhesiveness are complex and varied, highlighting the impact of food processing and preparation on food texture characteristics. In particular, the significant changes in cohesiveness at certain levels emphasize the need to consider the palatability and safety of food texture during preparation.

Additionally, this study utilized multiple linear regression analysis to quantitatively analyze the relationships between food adhesiveness, cohesiveness, and hardness and IDDSI levels, establishing a predictive model that can explain approximately 69.4% of the variance. This model effectively predicts the IDDSI of foods with different texture parameters. The results indicate that when formulating foods suitable for different IDDSI levels, it is essential to comprehensively consider the adhesiveness, cohesiveness, and hardness of the food. These findings not only deepen our understanding of the relationship between food texture characteristics and swallowing safety but also provide important support for decision-making in the food manufacturing and healthcare sectors. Future research should further explore the specific impact of different food components on texture parameters and how processing techniques can be improved to meet specific swallowing needs, achieving more precise control of food texture to better serve patient groups requiring specific IDDSI-level foods.

## Supporting information

**S1 Table. Hardness increase across different IDDSI levels and food categories.** This table displays the hardness increase and absolute increase observed when transitioning between IDDSI levels (0–7) for various food categories, including grains and tubers, vegetables, fruits, meats, eggs, and legumes. The data highlights the significant variations in hardness across levels, providing insights into textural changes critical for dysphagia management. (XLSX)

**S2 Table. Adhesiveness changes across different IDDSI levels and food categories.** This table summarizes the adhesiveness changes and absolute increase recorded during transitions between IDDSI levels (0–7) for different food categories. The results illustrate the varying degrees of adhesiveness changes, offering important implications for food preparation and

swallow safety.
(XLSX)

**S3 Table. Cohesiveness changes across IDDSI levels and food categories.** This table presents the cohesiveness changes and absolute increases measured across IDDSI levels (0–7) for multiple food categories. The findings reflect how cohesiveness varies between food types and levels, providing valuable information for developing texture-modified diets.
(XLSX)

**S4 Table. Statistical analysis of adhesiveness, hardness, and cohesiveness across IDDSI levels.** This table provides the results of statistical tests, including the W statistic and P-values, assessing the significance of differences in adhesiveness, hardness, and cohesiveness across different food samples and IDDSI levels. The analysis helps to evaluate the consistency and reliability of textural differences observed in the experimental data.
(XLSX)

## Author Contributions

**Conceptualization:** Muxi Chen.

**Data curation:** Muxi Chen, Yi Cheng, Mengyan Wang.

**Formal analysis:** Muxi Chen, Yi Cheng.

**Funding acquisition:** Lei Shi.

**Methodology:** Muxi Chen, Mengyan Wang.

**Project administration:** Muxi Chen, Yi Cheng.

**Resources:** Muxi Chen, Wen Hu, Lei Shi.

**Software:** Muxi Chen, Juan Duan.

**Supervision:** Muxi Chen, Wen Hu, Lei Shi.

**Validation:** Muxi Chen, Wen Hu.

**Visualization:** Muxi Chen, Juan Duan.

**Writing – original draft:** Muxi Chen, Yi Cheng, Mengyan Wang, Juan Duan.

**Writing – review & editing:** Yi Cheng, Wen Hu, Lei Shi.

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
