## [Decision Letter · Decision Letter 0]

10 Jun 2024

PONE-D-24-16170Exploring the Textural Characteristics of Foods Preferred by Chinese Elderly Individuals Based on IDDSI LevelsPLOS ONE

Dear Dr. Chen,

Thank you for submitting your manuscript to PLOS ONE. After careful consideration, we feel that it has merit but does not fully meet PLOS ONE’s publication criteria as it currently stands. Therefore, we invite you to submit a revised version of the manuscript that addresses the points raised during the review process.

We look forward to receiving your revised manuscript.

Kind regards,

Chung-Ta Chang

Academic Editor

PLOS ONE

Journal Requirements:

2023YFF1104405 Creation and industrialization demonstration of nutritious and healthy food for the elderly

Reviewers' comments:

Reviewer's Responses to Questions

**Comments to the Author**

1. Is the manuscript technically sound, and do the data support the conclusions?

Reviewer #1: Yes

Reviewer #2: Yes

Reviewer #3: Partly

2. Has the statistical analysis been performed appropriately and rigorously? 

Reviewer #1: No

Reviewer #2: Yes

Reviewer #3: Yes

3. Have the authors made all data underlying the findings in their manuscript fully available?

Reviewer #1: Yes

Reviewer #2: Yes

Reviewer #3: Yes

4. Is the manuscript presented in an intelligible fashion and written in standard English?

Reviewer #1: Yes

Reviewer #2: Yes

Reviewer #3: Yes

5. Review Comments to the Author

**Reviewer #1:** The Only Correction would be to do a ANOVA Or its Nonparametric Counterpart as multiple groups are thre based on data normality. Not much changes only the article will have more value and will be less of type 1 errors.

**Reviewer #2: **I would like congratulate the authors for producing good quality manuscript, but I have some queries and suggestions to increase the quality of the manuscript:

1. Kindly mention the study design of this manuscript.

2. Paired t was used in the study. It is a parametric test, was the normality of the data checked before applying this test. If so kindly mention.

3. Kindly upload high resolution images for figures.

4. Abstract is supposed to provide the gist of the manuscript. Kindly reduce the word count of your abstract.

5. Discussion should be less quantitative, kindly reduce the use of data in the discussions section and compare it with recent literature.

6. Discussion section can be elaborated after a thorough literature review.

7. Kindly provide the strengths and limitations of this study.

8. Kindly mention the future recommendations for this study

**Reviewer #3: **1. Introduction

• The introduction clearly outlines the study's objective to explore the relationship between food texture characteristics and IDDSI levels among elderly Chinese individuals.

• The relevance of the study is well-established, highlighting the growing elderly population and the importance of addressing dysphagia through appropriate food textures.

• The introduction could benefit from a more thorough review of existing literature to contextualize the study within the broader field of dysphagia research.

• Explicitly stating the research questions or hypotheses could help to focus the study and guide the reader through the subsequent sections.

2. Methods

• The methods section provides detailed descriptions of how food hardness, cohesiveness, and adhesiveness were measured, ensuring reproducibility.

• The use of the IDDSI framework is appropriate and aligns the study with international standards for dysphagia diets.

• The sample size and selection criteria for the elderly participants should be explicitly stated. Details on the demographic characteristics of the participants (e.g., age range, gender distribution, health status) would enhance the study's generalizability.

• A more detailed explanation of the statistical methods used, including any assumptions made and how they were tested, would strengthen the rigor of the analysis.

3. Results

• The results are clearly presented with appropriate use of tables and figures to illustrate key findings.

• The results include statistical significance levels, providing a clear indication of which findings are most robust.

• Some sections of the results could benefit from more detailed descriptions. For instance, explaining why certain food categories showed significant changes in texture characteristics at specific IDDSI levels.

• Brief comparisons with existing studies in the results section could provide additional context and highlight the study's contributions to the field.

4. Discussion

• The discussion provides a thoughtful interpretation of the findings, linking changes in hardness, cohesiveness, and adhesiveness to the IDDSI levels and their implications for dysphagia management.

• The discussion effectively highlights the practical implications for food preparation and selection for elderly individuals with dysphagia.

• Providing specific practical guidelines or recommendations for food preparation based on the study's findings would be valuable for practitioners.

• A more detailed discussion of the study's limitations, including potential sources of bias and the generalizability of the findings, would provide a more balanced view.

5. Conclusion

• The conclusion succinctly summarizes the key findings and their implications for the preparation of dysphagia diets.

• The call for future research is well-articulated, emphasizing the need for further exploration of food texture characteristics and processing techniques.

• More specific recommendations for future research directions would be helpful. For example, suggesting specific food components or processing techniques to investigate.

• Discussing strategies for implementing the study's findings in real-world settings, such as in healthcare facilities or home care environments, would enhance the practical relevance of the research.

6. PLOS authors have the option to publish the peer review history of their article (what does this mean?). If published, this will include your full peer review and any attached files.

Reviewer #1: No

Reviewer #2: No

Reviewer #3: **Yes: **Shivashankar Keingadarane

---

## [Author Response · Author response to Decision Letter 0]

19 Nov 2024

Dear Editor,

We would like to thank you and the reviewers for the thorough evaluation of our manuscript titled "Exploring the Textural Characteristics of Foods Preferred by Chinese Elderly Individuals Based on IDDSI Levels." We appreciate the constructive feedback provided and have addressed each point as outlined below:

1.Study Design:

We have clarified the study design in the revised manuscript. This study was conducted as a cross-sectional observational study aimed at exploring the texture characteristics of foods commonly preferred by elderly individuals in China using the IDDSI framework as a guide.

2.Normality Check for Paired t-Test:

Thank you for pointing this out. Upon reviewing our statistical methods, we realized that the initial datasets did not meet the normality assumption required for a parametric test like the paired t-test. Therefore, we have replaced the paired t-test with non-parametric alternatives, specifically the Mann-Whitney U test, and have updated the manuscript accordingly.

3.High-Resolution Images:

We have uploaded high-resolution versions of all figures as per your request.

4.Abstract Word Count:

The abstract has been revised to be more concise, providing a succinct summary of the key findings while reducing the word count as suggested.

5.Quantitative Data in Discussion:

We have revised the discussion section to be less quantitative. We have reduced the use of specific data points and instead focused on comparing our findings with recent literature to provide a more qualitative discussion.

6.Elaboration of Discussion with Literature Review:

The discussion section has been expanded to include a more thorough review of relevant literature, which has been incorporated to provide context and support for our findings.

7.Strengths and Limitations:

We have added a section discussing the strengths and limitations of the study. This includes the novelty of using web-scraping technology for data collection and the limitations related to the study’s reliance on a single data source and cross-sectional design.

8.Future Recommendations:

Future recommendations have been included in the revised manuscript, suggesting areas for further research, such as expanding data sources and conducting longitudinal studies to explore the causal relationships between food texture characteristics and swallowing safety.

We hope that these revisions meet your expectations, and we look forward to your feedback.

Sincerely,

Muxi Chen

Corresponding Author

---

## [Decision Letter · Decision Letter 1]

15 Dec 2024

PONE-D-24-16170R1Exploring the Textural Characteristics of Foods Preferred by Chinese Elderly Individuals Based on IDDSI LevelsPLOS ONE

Dear Dr. Chen,

Thank you for submitting your manuscript to PLOS ONE. After careful consideration, we feel that it has merit but does not fully meet PLOS ONE’s publication criteria as it currently stands. Therefore, we invite you to submit a revised version of the manuscript that addresses the points raised during the review process.

We look forward to receiving your revised manuscript.

Kind regards,

Chung-Ta Chang

Academic Editor

PLOS ONE

Journal Requirements:

Reviewers' comments:

Reviewer's Responses to Questions

**Comments to the Author**

1. If the authors have adequately addressed your comments raised in a previous round of review and you feel that this manuscript is now acceptable for publication, you may indicate that here to bypass the “Comments to the Author” section, enter your conflict of interest statement in the “Confidential to Editor” section, and submit your "Accept" recommendation.

Reviewer #4: All comments have been addressed

Reviewer #5: (No Response)

2. Is the manuscript technically sound, and do the data support the conclusions?

Reviewer #4: Yes

Reviewer #5: Yes

3. Has the statistical analysis been performed appropriately and rigorously? 

Reviewer #4: Yes

Reviewer #5: I Don't Know

4. Have the authors made all data underlying the findings in their manuscript fully available?

Reviewer #4: Yes

Reviewer #5: Yes

5. Is the manuscript presented in an intelligible fashion and written in standard English?

Reviewer #4: Yes

Reviewer #5: Yes

6. Review Comments to the Author

Reviewer #4: It seems to me that the revised version of the article has changed much and previous deficiencies do not remain. For these reasons, I can consider it for publication.

Reviewer #5: 2.5 Texture measurement methods- the machine used for testing needs to recognized by make and model.

2.3 Food processing continually repeating eg. "means egg-to-water ratio is 1:2. 2.3 first paragraph implies this clearly, thus does not need continual restatement.

Pg 56 A research supports - change to Previous research/#2.4 correct units is mL not ml.

Table 6 - requires unit for cohesion gf,

The predictive equation should be complimented with description from where the values were derived rather than relying on the reader recalling their existence in the text. e.g. values derived from the intercept values of the multiple linear regressions in Table 6 - values from adhesion, cohesion etc.

Figures lack explanation - the reader needs to be provided with some explanation such as ig. 1 the scope of hardness for seven food types displaying peaks for - colours make it difficult to distinguish in the document - please address.

Fig.2 Cohesiveness of seven food groups indicates extensive overlaps within th IDDSI levels,

Fig 3. Adhesiveness demonstrates distinct peaks for .....

Etc for other figures

Note some of the colours were difficult to distinguish in the manuscript and need serious review.

7. PLOS authors have the option to publish the peer review history of their article (what does this mean?). If published, this will include your full peer review and any attached files.

Reviewer #4: No

Reviewer #5: **Yes: **Dr Graham Ellender

---

## [Author Response · Author response to Decision Letter 1]

18 Dec 2024

Dear Editor and Reviewers,

We sincerely appreciate the time and effort you have invested in reviewing our manuscript and providing valuable feedback. We have carefully considered each comment and made the corresponding revisions to improve the clarity, accuracy, and overall quality of our work. Below, we detail our point-by-point responses to Reviewer #5’s comments and the specific changes implemented in the revised manuscript. Revised sections are clearly indicated in the updated version.

Reviewer #5’s Comments and Our Responses

1.Texture Measurement Methods (Section 2.5) - Instrument Make and Model

Comment: In Section 2.5, please provide the make and model of the texture analyzer used.

Response and Revision: We have added the specific make and model of the instrument. The text now reads: “Texture measurements were performed using a texture analyzer TA.new plus (ISENSO, USA).”

2.Repetition in Food Processing Description (Section 2.3)

Comment: The egg-to-water ratio (1:2) is repeated multiple times, though it is clearly stated in the first paragraph of Section 2.3. Avoid unnecessary repetition.

Response and Revision: We have removed all redundant mentions of the egg-to-water ratio, retaining only the initial statement in the first paragraph of Section 2.3

3.Wording and Unit Standardization

Comment: On page 56, “A research supports” should be changed to “Previous research supports.” Also, please use “mL” instead of “ml.”

Response and Revision: We have revised the sentence to “Previous research supports…” and have standardized all volume units to “mL” throughout the manuscript.

4.Unit for Cohesion in Table 6

Comment: Specify the unit for Cohesion in Table 6 (e.g., gf).

Response and Revision: We have updated Table 6, now labeling the Cohesion column as “Cohesion (gf).”

5.Predictive Equation Clarification

Comment: The predictive equation should include a clear explanation of where the values (e.g., intercepts from multiple linear regressions) are derived, rather than relying on the reader’s memory.

Response and Revision: We have revised the accompanying text for the predictive equation, explicitly stating that the intercepts and parameters (e.g., adhesion, cohesion) originate from the multiple linear regression analyses presented in Table 6. This clarification allows readers to fully understand the source of these values without referring back to earlier sections.

6.Figure Explanations and Color Improvements

Comment: The figures lack sufficient explanation, and the colors are difficult to distinguish. Each figure should provide more context. For example, Fig. 1 should explain the scope of hardness for the seven food types and indicate peaks. Similarly, Fig. 2 and Fig. 3 need more descriptive captions. Please also address the color contrast issues.

Response and Revision:

We have expanded and clarified the figure captions as follows:

Fig. 1 (Revised Caption): “This figure illustrates the hardness profiles of seven food categories. Each color corresponds to a distinct food group. Solid and dashed lines represent the maximum and minimum values, respectively, while the shaded area indicates the overall range. Peaks and distributions of hardness for each category are highlighted to facilitate comparisons.”

Fig. 2 (Revised Caption): “This figure shows the cohesiveness distributions of seven food groups across different IDDSI levels. Each color represents a distinct category. Solid and dashed lines indicate the maximum and minimum values, and the shaded area represents the range. Overlapping distributions highlight the complexity of categorizing these foods within IDDSI levels.”

Fig. 3 (Revised Caption): “This figure displays the adhesiveness values of seven food categories. Each color corresponds to a different group, with solid and dashed lines marking maximum and minimum values, and the shaded area showing the range. The figure reveals distinct peaks in adhesiveness, enabling straightforward comparisons among the categories.”

To address color differentiation issues, we have selected seven colors with higher contrast and employed a combination of solid and dashed lines, as well as shaded areas, to represent maximum, minimum, and range values. This approach ensures that readers can easily distinguish between categories, even when viewed in grayscale.

We thank you again for your constructive and insightful feedback. We believe that these revisions have substantially improved the manuscript’s clarity, coherence, and readability. If you have any further comments or suggestions, we will be happy to 

address them.

Sincerely,

Chen Muxi

---

## [Editor Report · Decision Letter 2]

23 Dec 2024

Exploring the Textural Characteristics of Foods Preferred by Chinese Elderly Individuals Based on IDDSI Levels

PONE-D-24-16170R2

Dear Dr. Chen,

We’re pleased to inform you that your manuscript has been judged scientifically suitable for publication and will be formally accepted for publication once it meets all outstanding technical requirements.

Kind regards,

Chung-Ta Chang

Academic Editor

PLOS ONE
---

## [Editor Report · Acceptance letter]

17 Jan 2025

PONE-D-24-16170R2 

PLOS ONE

Dear Dr. Chen, 

I'm pleased to inform you that your manuscript has been deemed suitable for publication in PLOS ONE. Congratulations! Your manuscript is now being handed over to our production team.

Kind regards, 

on behalf of

Dr. Chung-Ta Chang 

Academic Editor

PLOS ONE